# Monitoring Expression of Balance during Therapy in Children with Postural Disorders

**DOI:** 10.3390/children10060974

**Published:** 2023-05-30

**Authors:** Arkadiusz Żurawski, Zbigniew Śliwiński, Dorota Kozieł, Wojciech Kiebzak

**Affiliations:** Collegium Medicum, Jan Kochanowski University in Kielce, 25-369 Kielce, Poland

**Keywords:** spine disorders, equivalent reactions, pedobarographic platform, neurophysiological methods

## Abstract

In the literature, we find information about the impact of changes in the shape of the spine on the efficiency of equivalent reactions, and we also find information about the methods and effects of physiotherapy in improving equivalent reactions. However, there is a lack of publications showing the process of changes taking place over time and defining their nature at individual stages of treatment. The aim of this study is to present the process of monitoring the expression of balance in patients aged 8–12 years with postural disorders, in the course of the therapeutic process. The shape of the spine and the efficiency of equilibrium reactions in standing posture and during gait were assessed in all the subjects. The subjects were put into two groups: with a shape-of-spine disorder and without one. The recommended set of therapeutic activities in home conditions lasted about 20 min and was performed by the child with a parent’s supervision. The therapeutic program was based on elements of neurophysiological methods: Vojta and PNF. The following parameters were measured: the maximum movement of the center of pressure (CoP) in the frontal plane during gait, the maximum movement of the CoP in the sagittal plane, the movement of the CoP in the frontal plane in static conditions and the movement of the CoP in the sagittal plane in static conditions. Six statistically significant differences were recorded, and all of them were related to measurement I. The Friedman test result was statistically significant for all the indexes. Post hoc analyses were performed using the Dunn–Bonferroni test. The children with shape-of-spine disorders had a lower efficiency of equivalent reactions in relation to the children without disorders. The therapy with the application of neurophysiological methods in the treatment of shape-of-spine disorders improved equilibrium reactions in these patients. Long-lasting and thorough observations of the therapeutic process in children with shape-of-spine disorders should include the monitoring of the efficiency of equilibrium reactions.

## 1. Introduction

An overview of the literature and our own clinical observations point to an interesting aspect of the issues related to the shape of the spine. A correct shape of the spine allows one, for example, to keep the eyesight directed horizontally, in accordance with the guidelines of otoscopy, in order to maintain the eyesight in the transverse plane and to limit the movement of the center of gravity to the plane of support [1,2]. It was found that children with shape-of-spine disorders are characterized by balance disorders, weakened response to stimuli and a longer time of return to the starting position [3].

The abovementioned balance is a general term describing changes in the alignment of particular parts of the body against one another and in relation to space [4]. It is an important part of the neuromuscular control of the mechanism preventing falls. It is a state characterized by the vertical orientation of the body, maintained due to the balance of forces working on it [4]. It allows one to regain the original positioning of the body while performing various motor tasks and after their completion [4]. The ability to maintain good, stable posture is a prerequisite for performing the majority of purposeful movements, including gait [5].

The efficiency of equilibrium reactions can be assessed by observing the movements of the center of pressure (CoP), which is the point of application of the ground reaction force vector. It is the weighted mean of all pressures on the surface in contact with the ground. Since the spontaneous movement of the CoP is two-dimensional information, by analyzing the individual elements of a statokinesiogram path, i.e., separate sway in the sagittal plane and in the frontal plane, they can be used to determine the efficiency of equilibrium reactions [5,6,7]. Statokinesiogram depicts the movement of the pressure center in a coordinate system in which the axis x refers to swinging in the direction right–left, and the axis y refers to swinging in the direction forward–backward.

It has been found that shape-of-spine defects are one of the causes of dysfunction in the reflex maintenance of static equilibrium, as well as the possibility of corrective reflexes overlapping the free movement program [4]. Therefore, shaping the postural reflex is not only related to the muscular system but also to improving its function in adjustment and equilibrium reactions. Along with the increase in the dysfunction of the shape of the spine, the extension of the CoP path in a given time unit is observed, which means that the equilibrium reactions get weakened [4].

There is an interesting dependency between the shape of the spine and the efficiency of equilibrium reactions during gait [8]. In the case of minor shape-of-spine defects, this dependency mainly concerns the sagittal plane; clinical observations of visible postural disorders indicate that in these cases, the abovementioned dysfunctions also include the movement of the center of gravity in the frontal plane [9]. The shape of the spine is related to the ergonomics of gait, and its proper values are necessary for the correct movement of the center of gravity during the person’s movement [4].

It is important to improve equilibrium reactions in order to improve the quality of life [10]. The process of improving balance disorders is based on the mechanism of neuroplasticity of the central nervous system, as a result of the processes of adaptation, habituation and substitution [11]. The aim of the therapy is to improve the visual-vestibular interaction during head movements and to stimulate proprioceptive functions, which translates into a reduction in balance disorders [12]. The success of the treatment process depends mainly on the systematic implementation of a personalized treatment plan [13,14], and this applies to the therapy of both shape-of-spine disorders and balance disorders.

The authors hypothesized that by observing CoP deflections, we are able to determine the level of neuromuscular adaptation to changes in the shape of the spine. Observing the decrease in CoP deflections, we can assume that the body adapts to changes in the musculoskeletal system.

In the literature, we find information about the impact of changes in the shape of the spine on the efficiency of equivalent reactions [15,16], and we also find information about the methods and effects of physiotherapy in improving equivalent reactions [16]. However, there is a lack of publications showing the process of changes taking place over time and defining their nature at individual stages of treatment. The present study aims to present the process of monitoring the changes in the expression of balance in patients aged 8–12 years with postural disorders, in the course of the therapeutic process.

## 2. Materials and Methods

### 2.1. Subjects

Patients (*n* = 312) aged 8–12 years were evaluated. All participants were divided into 2 groups.

First group (study)—children with a recognized shape of the spine defect. Children from group 1 underwent a four-month therapy conducted under the supervision of a physiotherapist.

Group 2 (control)—children without spinal shape defects.

Group 1 consisted of 211 people (53% women, 46.92% men) aged 10.72 years (SD = 1.25). The characteristics of the respondents in this group were as follows:-Height 1.4 m (SD = 0.16);-Body weight 34.7 kg (SD = 11.84);-BMI 20.15 (SD = 2.35).

Group 2 consisted of 101 people (50.5% women, 49.5 % men) aged 10.69 (SD =1.44). The characteristics of the respondents in this group were as follows:-Height 1.39 m (SD = 0.13);-Body weight 38.69 (SD = 8.03);-BMI 20.02 (SD = 2.52).

Inclusion criteria:-Age 8–12 years;-Diagnosed shape-of-spine defect;-The legal guardian’s consent for the minor’s participation in the study.

Exclusion criteria:-The presence of comorbidities that may affect the shape-of-spine disorder;-Interruption or non-compliance with the orders included in the therapeutic procedure;-BMI below the 10th and above the 90th percentile.

Participation in the study was voluntary, with ensured anonymity in accordance with Ustawa o ochronie danych osobowych of 29.08.1997 (Dz.U.Nr 133 item 883) [17].

### 2.2. Study Project

The shape of spine and the efficiency of equilibrium reactions in standing posture and during gait were assessed in all the subjects. The individual elements of the statokinesiogram path were analyzed in sagittal and frontal planes, which allows one to objectively determine the efficiency of equilibrium reactions [5,6,7]. The subjects were put into 2 groups: with a shape-of-spine disorder and without one. Based on the functional assessment and the Diers computer analysis (Figure 1), a therapeutic program was fitted for the children in the study group. The DIERS Formetric is a light optical scanning method based on video rasterstereography (VRS). Accordingly, the system consists of a light projector that projects a line grid on the back of the patient which is recorded by an imaging unit. A computer software program analyzes the line curvature and generates from it—by means of the method of photogrammetry—a three-dimensional model of the surface, comparable to a plaster cast. The program assumed that the child would perform preordered therapeutic activities at home twice a day, supervised by the parent, taking care of the proper correction of body posture with the use of a mirror.

The study conduct was approved by The Bioethics Committee, Approval No. 1/2016, issued on 15 January 2016.

### 2.3. Intervention

The therapeutic protocol was performed by the child under the supervision or, if necessary, with the help of a parent at home. The parents were trained by a physiotherapist in the recommended techniques. The duration of exercises performed at home was 20 min. The therapeutic program consisted of elements of neurophysiological methods: PNF and Vojta. Vojta method focuses on the control of the cervical–thoracic and thoracic–lumbar spine transition to achieve a symmetrical position of the shoulder and pelvic girdles, concentric activity of the abdominal muscles, activity of autochthonous muscles and normal muscle activity within the hip joint. Patients reported to the physiotherapist every four weeks. During each visit, the analyzed parameters were measured, and a functional test was performed, on the basis of which the therapy was selected (Figure 2). The protocol of the presented study assumed recording the results of four consecutive visits.

Analysis of ground reaction forces and equilibrium reactions. In group 1, the measurements were made four times at four-week intervals, and in group 2, the analyzed parameters were measured once.

During the examination, the efficiency of equilibrium reactions during gait was analyzed. For this purpose, DIERS Pedogait was used, consisting of a treadmill and a pedobarographic platform built into it [14]. The DIERS Pedogait system allows the functional representation of the foot pressure reaction forces while walking. The integrated measuring platform is 1.0 m long with 5.376 sensors for an exact capture of the pressure values. The admission frequency amounts to 100 Hz which corresponds to a tact frequency of 10 ms. Thereby, false measurements and artifacts are avoided.

The standing treadmill can be used for static measurements of foot pressure as well as for stabilometry [18]. The measurements were made while walking slowly at a speed of 2 km/h. The following parameters were measured:Maximum movement of the CoP in the frontal plane during gait—the parameter calculated in centimeters is the biggest CoP displacement in the frontal plane during gait over a distance of 16 m.Maximum movement of the CoP in the sagittal plane—the parameter calculated in centimeters is the biggest CoP displacement in the sagittal plane during gait over a distance of 16 m.

Measurements of the movement of the CoP in a static position were made with the DIERS Pedoscan [14]. The examined person stood on the 80 × 100 cm platform, in its central part. The feet were placed forward in their natural relaxed position.

The following parameters were measured:Movement of the CoP in the frontal plane in static conditions—the parameter was calculated in centimeters, showing the biggest center-of-pressure displacement of the body onto the ground during 10 s in the frontal plane (to the left and right).Movement of the CoP in the sagittal plane in static conditions—the parameter was calculated in centimeters, exemplifying the biggest center of pressure of the body onto the ground during 10 s in the sagittal plane (forward and backward).

### 2.4. Statistical Methods Used

Statistical analyses were carried out using the Statistica program (IBM SPSS Statistics25). Descriptive statistics were analyzed: mean (M) and standard deviation (SD).

The evaluation of the normality of the distribution of variables was performed with the non-parametric Kolmogorov–Smirnov test [19].

The Mann–Whitney U test (U) was used to assess the significance of differences between parameters in the study and control groups. This test was chosen due to the fact that most of the analyzed variables had a distribution other than normal [20].

The test statistic is denoted as U and is the smaller of U_1_ and U_2_, as defined below:U_1_ = n_1_n_2_ + n_1_(n_1_ + 1)/2 − R_1_
U_2_ = n_1_n_2_ + n_2_(n_2_ + 1)/2 − R_2_
where n_1_ and n_2_ are the sample sizes for samples 1 and 2, respectively, and R_1_ and R_2_ are the sum of the ranks for samples 1 and 2, respectively.

Friedman’s test [21] was used to analyze changes occurring between successive measurements in the study group. This test allows one to determine the differences between four measurements simultaneously. The Friedman test is the non-parametric alternative to the one-way ANOVA with repeated measures. It is used to test for differences between groups when the dependent variable being measured is ordinal. It can also be used for continuous data that have violated the assumptions necessary to run the one-way ANOVA with repeated measures.

Friedman’s test assumes a model of the form
*xijk* = *μ* + *αi* + *βj* + *εijk*
where *μ* is an overall location parameter, *αi* represents the column effect, *βj* represents the row effect, and *εijk* represents the error. This test ranks the data within each level of B and tests for a difference across levels of A. What follows is that the *p* value for the null hypothesis that pfriedman *αi* = 0. If the *p* value is near zero, this casts doubt on the null hypothesis. A sufficiently small *p* value suggests that at least one column-sample median is significantly different from the others; i.e., there is a main effect due to factor A.

Statistically significant results of the Friedman test were subjected to post hoc tests with the Dunn–Bonferroni test [χ2] to counteract the multiple-comparison problem of reducing the nominal significance level of each set of related tests [22]. The Bonferroni correction modifies probability (*p*) values to account for the increased likelihood of a “type I” error while performing numerous statistical tests. The formula to calculate the z-test statistic for the difference between two groups is
z_i_ = y_i_/σ_i_
where *i* is one of the 1 to *m* comparisons, y_i_ = W_A_ − W_B_ (where W_A_ is the average of the sum of the ranks for the ith group), and σ_i_ is calculated as follows:σ_i_ = √((N(N + 1)/12) − (ΣT^3^_s_ − T_s_/(12(N − 1))/((1/n_A_) + (1/n_B_))
where N is the total number of observations across all groups, *r* is the number of tied ranks, and T_s_ is the number of observations tied at the *s*th specific tied value.

## 3. Results

Basic descriptive statistics of the studied quantitative variables were calculated along with the Kolmogorov–Smirnov test checking the normality of the distribution of these variables. The analyses were performed separately for the study and the control groups. The vast majority of measurements had distributions different from the normal ones. In this work, statistical analyses were performed with non-parametric tests in order to keep the consistency of the scores.

### 3.1. Level of Efficiency of Equilibrium Reactions in the Test and Control Groups

The level of the efficiency of equilibrium reactions in the test and control groups was verified. A series of analyses that compared the two groups was performed with the Mann–Whitney U test. Due to one-time measuremsent in the control group, the scores obtained in the study group in measurements I and IV were compared to the same measurement in the control group, and then the strength of the observed effects was compared.

Six statistically significant differences were recorded, and all of them were related to measurement I. The scores for the variables of the maximum movement of the CoP in the frontal plane and the maximum movement of the CoP in the sagittal plane during gait, the maximum movement of the CoP to the left and right statically and the maximum forward and backward movement of the CoP were statistically higher in the study group. The fact that the differences in measurement IV did not even come close to statistical significance indicates a significant reduction in the differences between the groups as a result of the therapy. The detailed data are presented in Table 1.

### 3.2. Changes Occurring between Subsequent Measurements in the Study Group

The criteria for the level of the center-of-pressure displacement in static and dynamic conditions were analyzed in terms of changes occurring between subsequent measurements in the study group. A series of non-parametric Friedman analyses of variance was performed, and they were subjected to post hoc tests with the Dunn–Bonferroni test (Table 2).

Friedman’s test result was statistically significant for all indicators. Post hoc analyses were performed using the Dunn–Bonferroni test. In terms of the index, the maximum displacement of the center of gravity in the frontal plane during gait was recorded at the lowest level in measurement IV. This result was statistically significantly lower than in measurements I and II. Measurement III assumed intermediate values. In terms of the indicator of the maximum displacement of the center of gravity in the sagittal plane during gait, the lowest score was presented in measurement IV. It was statistically significantly lower than in measurements I, II and III. There was also a statistically significant difference in the level of this indicator between measurements I and II, and a difference close to statistical significance was noted between measurements I and III. The maximum shift of the center of gravity to the left was the smallest in measurement IV, and the difference was statistically significant. The maximum shift of the center of gravity to the right and the maximum shift of the center of gravity forward presented similar results—a visible decrease in the level of these variables in measurement IV and significant differences between measurements I, II and III. In terms of the maximum rearward displacement of the center of gravity, the results were similar, i.e., the lowest result in measurement IV differed statistically significantly from the level in measurements I, II and III; additionally, a statistically significant difference between measurement I and measurements II and III was recorded. Detailed data are shown in Table 2.

## 4. Discussion

The shape of the spine is susceptible to the influence of many factors, due to the complex system of its control [23]. The ability to keep a correct, stable posture is a prerequisite for most purposeful movements, including gait [5]. Therefore, monitoring the treatment process of patients with shape-of-spine disorders should take into account all the dysfunctions developing in their body [24], including dysfunctions in the efficiency of equilibrium reactions.

Reliable monitoring of the treatment process enables the choice of a suitable, personalized rehabilitation program that gives a chance to achieve positive effects in the form of improving equilibrium reactions in patients with shape-of-spine defects [13,14]. The rehabilitation techniques used in our study affected the efficiency of equilibrium reactions in the children subjected to therapy. The parameters describing equilibrium responses assessed during gait improved. The maximum center-of-pressure displacement in the sagittal plane decreased by 3.82 cm (19%), and the maximum center-of-pressure displacement in the frontal plane decreased by 0.66 cm (6%). The static evaluation showed a significant reduction in the movement of the center of gravity to the left by 0.42 cm (41%), to the right by 0.25 cm (31%), forward by 0.33 cm (28%) and backward by 0.25 cm (36%). During the analysis of the scores, a bigger shift of the center of gravity to the left, as well as a bigger effect of improving this parameter in relation to the right side, is clearly visible. Similar observations were made by Bruyneel who analyzed the balance in a sitting position in the patients with spine deviation; this dependency is explained by the more frequent occurrence of deviation with concavity on the left side [25], which is reflected in the results obtained by the authors (125 subjects had a dominant concavity on the left side and 86 on the right).

It is worth noting that after 4 months of rehabilitation, all the parameters improved, reaching the values typical of the control population. No deterioration from baseline values was registered. The improvement in the analyzed parameters trended from the second examination on, so the effect of the therapy on the values of the equilibrium parameters began at the latest after a month of its implementation, although the biggest effect was observed between the third and fourth appointments.

An analysis of the literature showed a small number of research papers describing the effect of the shape-of-spine correction on the efficiency of equilibrium reactions. Attention should be drawn to the work by Kinel et al., which describes the improvement of equilibrium reactions and motor coordination after using the Vojta method in patients with postural disorders [13]. During a six-week program of rehabilitation with PNF, Lee noted a decrease in the movement of the center of gravity in both static and dynamic examination, and the parameters describing the positioning of the spine in space also improved [14]. Winter [4] also points out that the improvement in balance can be expected along the improvement in the shape of the spine. This involves a shift in the center of mass in the upper torso, which is then shifted to the ground through the hip and ankle joints. Rougier [26] explains this phenomenon by the fact that the displacements change through the forces shifted to the feet and are partially reduced by the compensations occurring in the abovementioned joints. This causes a change in the load on the feet and disturbs afferent information, which affects the incorrect perception of one’s own body in space, as indicated by Roll and Kavounoudias [27,28]. Carlsöö, on the other hand, attributes the improvement in the efficiency of equilibrium reactions resulting from the restoration of the physiological axis of the body to the reduction in tensions coming through the sacro-dorsal and calf muscles to the feet [29].

Based on the above findings, it can be concluded that the efficiency of equilibrium reactions is an important aspect of the process of shape-of-spine correction. Naulut points out that it is the size of the deformity of the spine that determines the size of the equilibrium reaction disorders [9]. The quality and quantity of stimuli determine mutual dependencies between the dynamic and static parts of the musculoskeletal system, which are important components of the balance system [30,31,32,33].

As many as five out of six parameters determining the efficiency of the equilibrium reaction did not improve immediately, and even deterioration was observed. In the literature, we found that it is the size of postural dysfunctions that determines the size of the disturbances of equivalent reactions, as reported by Naulut [9]. Winter [4] also points out that with the restoration of the physiological position of the spine, an improvement in balance can be expected. It involves a shift in the center of mass in the upper torso, which is then transferred to the ground through the hip and ankle. Carlsöö, on the other hand, attributes the improvement in balance resulting from the restoration of the physiological size of thoracic kyphosis to a reduction in tension flowing through the sacro-dorsal and calf muscles to the feet [29]. The lack of immediate improvement after the implementation of therapy restoring the correct shape of the spine may be the result of compensation in the form of torso tilt or changes in the angular position of the joints of the lower limbs or foot loads, as reported by Rougier [26] and Suoza et al. [34]. This causes a change in the load on the feet and the disturbance of afferent information, which affects the incorrect perception of one’s own body in space, as indicated by Roll and Kavounoudias [27,28]. This means that the brain perceives the body as vertical when it is tilted laterally. The process of improving balance disorders is based on the mechanism of neuroplasticity of the central nervous system, as a result of the processes of adaptation, habituation and substitution [11]. The observation of the subsequent results presented in the study indicates that this adaptation of the central nervous system to new parameters of body posture may take up to 4 months. However, this thesis requires verification in other groups of patients and with the use of other therapeutic methods.

When planning the treatment process, actions aimed at improving the ability to maintain the body in space should be considered [35]. The disorder of the physiological curves of the spine will always have a negative effect on balance, which should be taken into account when planning the treatment process.

## 5. Conclusions

The crucial issue in the therapeutic process is examination, observation and therapy. This research confirms that children with shape-of-spine disorders are characterized by a lower efficiency of equivalent reactions in relation to children without disorders. Therefore, the therapeutic process in children with shape-of-spine disorders should include the monitoring of the efficiency of equilibrium reactions. What is more important, the observation should be long-lasting and thorough. In the presented study, a clear decrease in CoP deviations in the study group was observed only after twelve weeks.

## 6. Limitations

The study protocol did not include angular measurements in the lower limb joints nor the activity of postural muscles, and these are important components of the process of controlling the positioning of the body in space. In this work, only one of the components of the entire cause-and-effect chain aimed at keeping the body in balance was analyzed. An important limitation of the presented study is the lack of a priori power analysis; therefore, it is not known whether the conclusions can be transferred to the entire population.

This study assumes a description of how to monitor changes in a certain phenomenon. Continuous measurements were not used in the control group, and therefore it is impossible to fully rule out changes in the analyzed parameters as a result of puberty. The impact of growth on the analyzed parameters should not be high (four-month observation), but it may occur.

## 7. Clinical Implications

The results presented in this work explicitly show that the efficiency of equilibrium reactions and the shape of the spine are interrelated. Planning the treatment, both in the case of shape-of-spine disorders and deficits in equilibrium reactions, cannot focus on just one of these areas but must be comprehensive.

All modern guidelines for the treatment of spinal deformities assume thorough postural re-education. The balance and tilt of the CoP are excellent tools for monitoring the efficiency of maintaining a corrected body posture and can be included in the diagnostic process of these dysfunctions.

## Figures and Tables

**Figure 1 children-10-00974-f001:**
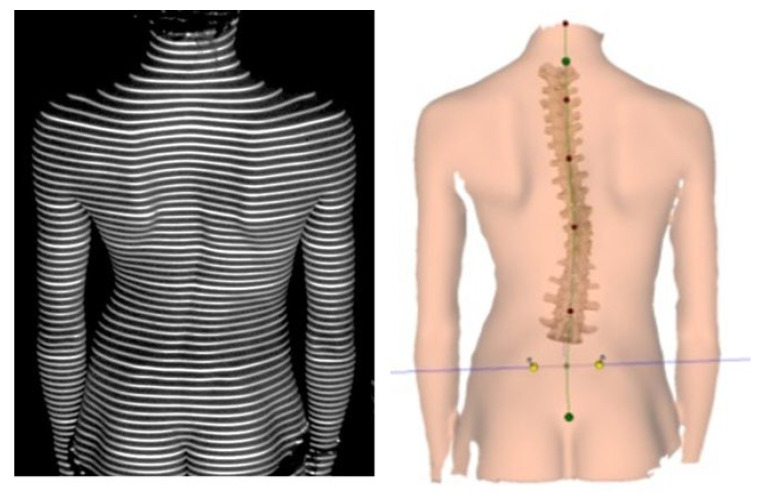
One of the types of spinal shape disorders—right-convex curvature, in the back view and the image of the Diers system.

**Figure 2 children-10-00974-f002:**
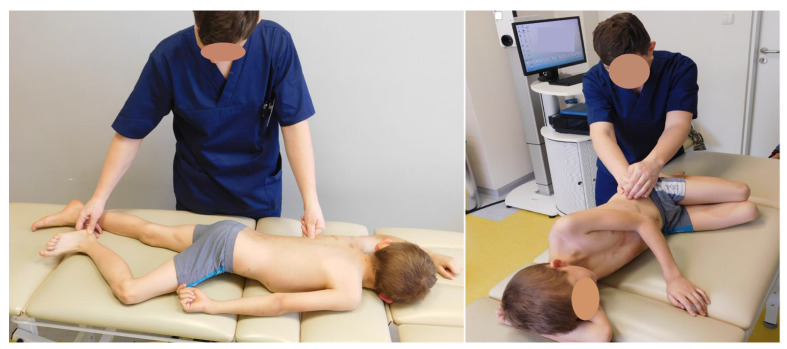
Exemplary techniques of working with the patient. The Vojta method element on the left and the PNF method element on the right.

**Table 1 children-10-00974-t001:** Comparison of the study and control groups in terms of the level-of-pressure distributions and the center of gravity in static conditions and during gait.

		Study Group		Control Group					
	Measurement	*M*	*SD*	*M*	*SD*	*U*	*Z*	*p*	*r*
Maximum movement of the CoP in the frontal plane—gait (cm)	I	10.46	2.50	9.87	1.92	9126.5	−2.051	0.040	0.12
	IV	9.80	1.94	9.87	1.92	10,650.5	−0.007	0.995	0.00
Maximum movement of the CoP in the sagittal plane—gait (cm)	I	20.20	5.83	16.03	3.23	5542.5	−6.858	<0.001	0.39
	IV	16.38	3.68	16.03	3.23	10,198.0	−0.614	0.539	0.03
Maximum movement of the CoP to the left—static (cm)	I	1.03	1.55	0.56	0.33	6846.5	−5.109	<0.001	0.29
	IV	0.61	0.41	0.56	0.33	9964.0	−0.928	0.354	0.05
Maximum movement of the CoP to the right—static (cm)	I	0.80	0.64	0.55	0.35	7445.0	−4.307	<0.001	0.24
	IV	0.55	0.36	0.55	0.35	10,577.5	−0.105	0.917	0.01
Maximum forward movement of the CoP—static (cm)	I	1.18	1.01	0.83	0.41	7108.5	−4.758	<0.001	0.27
	IV	0.85	0.44	0.83	0.41	10,273.0	−0.513	0.608	0.03
Maximum backward movement of the CoP—static (cm)	I	0.69	0.97	0.43	0.28	8145.5	−3.367	0.001	0.19
	IV	0.44	0.36	0.43	0.28	10,587.5	−0.091	0.927	0.01

*M*—mean; *SD*—standard deviation; *U*—U Mann–Whitney’s test result; *Z*—standardized value; *p*—statistical significance; *r*—effect size.

**Table 2 children-10-00974-t002:** Changes during therapy in the level-of-pressure distributions and center of gravity in static conditions and during gait. Statistically significant differences at the level of *p* < 0.05 were marked with different letter indexes. The Dunn–Bonferroni test.

		Measurement	*M*	*SD*	
Maximum movement of CoP in the frontal plane—gait (cm)		I	10.46 a	2.50	
		II	10.38 a	2.51	χ^2^(3) = 9.67
		III	10.15 ab	2.30	*p =* 0.022
		IV	9.80 b	1.94	
Maximum movement of the CoP in the sagittal plane—gait (cm)		I	20.20 a	5.83	
		II	18.66 b	5.78	χ^2^(3) = 42.60
		III	18.96 ab	4.94	*p <* 0.001
		IV	16.38 c	3.68	
Maximum movement of the CoP to the left—static (cm)		I	1.03 a	1.55	
		II	1.02 a	1.26	χ^2^(3) = 38.21
		III	1.05 a	1.73	*p <* 0.001
		IV	0.61 b	0.41	
Maximum movement of the CoP to the right—static (cm)		I	0.80 a	0.64	
		II	0.89 a	1.35	χ^2^(3) = 27.11
		III	0.92 a	1.16	*p <* 0.001
		IV	0.55 b	0.36	
Maximum forward movement of the CoP—static (cm)		I	1.18 a	1.01	
		II	1.18 a	1.03	χ^2^(3) = 44.41
		III	1.28 a	1.28	*p <* 0.001
		IV	0.85 b	0.44	
Maximum backward movement of the CoP—static (cm)		I	0.69 a	0.97	
		II	0.65 b	0.96	χ^2^(3) = 474.11
		III	0.67 b	1.14	*p <* 0.001
		IV	0.44 c	0.36	

*M*—mean; *SD*—standard deviation.

## Data Availability

The data presented in this study are available on request from the corresponding author.

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
