# Peer review of "Monitoring Expression of Balance during Therapy in Children with Postural Disorders"

_children, 2023, doi:10.3390/children10060974_

Round 1

Reviewer 1 Report

The paper presents a study on monitoring the efficiency of equilibrium reactions during therapy in children with postural disorders.

The manuscript is well written and understandable.

The research topic is of relevance for both research and practice.

Important strengths include the detailed monitoring assessments.

Has an a priori power analysis been conducted? Are estimates reliable with this sample?

Are results generalizable? Do findings transfer to other populations, contexts, etc.?

One conclusion is that “The therapy with the application of neurophysiological methods in the treatment of shape of spine disorders improves equilibrium reactions in these patients.”

Speaking about intervention effects, a control therapy needs to be applied; a significant time x therapy versions interaction needs to be observed.

If this approach has not been done, please discuss as limitation and revise the conclusion.

The conceptual implications can be strengthened. Which particular health theories are advanced? How?

The practical implications for children in real life functioning can be highlighted with more detailed examples from everyday life.

Do the authors assume differences between boys versus girls?

Do they assume differences between different age groups reflecting different developmental stages?

Author Response

Dear Reviewer,
Thank you very much for your time and valuable comments. I will address them in turn below.

  1. Has an a priori power analysis been conducted? Are estimates reliable with this sample? Are results generalizable? Do findings transfer to other populations, contexts, etc.?

         Unfortunately, no a priori analysis was performed before the study, so the presented results cannot be applied to the entire population with certainty. We have added such an explanation in the restrictions section.

           2. One conclusion is that “The therapy with the application of neurophysiological methods in the treatment of shape of spine disorders improves equilibrium reactions in these patients.”

Speaking about intervention effects, a control therapy needs to be applied; a significant time x therapy versions interaction needs to be observed.

If this approach has not been done, please discuss as limitation and revise the conclusion.

Thank you very much for this attention. We have analyzed the text again and it is absolutely correct. So we decided to remove this request. While we as clinicians know it to be true, it does not follow from the results presented.

3. The conceptual implications can be strengthened. Which particular health theories are advanced? How?

The practical implications for children in real life functioning can be highlighted with more detailed examples from everyday life.

Thank you for this suggestion. We have expanded the clinical implications section.

4. Do the authors assume differences between boys versus girls?

At this age, before puberty, these differences do not yet occur. We have defined this in another publication (https://doi.org/10.3390/ijerph192316287), but these issues have not been addressed in the text presented here.

5. Do they assume differences between different age groups reflecting different developmental stages?

Yes, there are definitely differences between age groups. Balance develops rapidly at the age of 6-9 years, then deteriorates during puberty and rapid growth, to reach its target fitness after about 19 years of age. For this reason, the age group after the period of rapid development of balance and before puberty was selected for the study. This age is also clinically justified, because this is when the highest percentage of scoliosis has its onset.

Thank you again for your comments. We hope that the text in its current form will meet the requirements.

Reviewer 2 Report

Thanks for the good research. When reviewing the entire paper, a few corrections and supplements will improve the quality of the research.

Author Response

Dear Reviewer,
Thank you very much for your time and valuable comments. I will address them in turn below.

1. [Line 37] It would be nice to have an example of the shape of spine disorders.

Thank you for your suggestion. Appropriate figure added.

2. There seems to be a lack of research necessity. To describe in detail the research necessary for research purposes.

A proper explanation has been added in the introduction.

3. [Line 94] It would be nice to have an example of the diagnosed shape of a spine defect.

In the introduction, by adding a figure with a photograph of the spinal shape disorder, it was combined with a diagnostic image.

4. It would be nice to present a table with the subject’s characteristics.

Data on the characteristics of the study and control groups are included in the "Subjects" section of the "Material and Methods" chapter.

5. It would be nice to present a table or figure with the therapeutic activities such as PNF and Vojta methods

Thank you for this suggestion. The corresponding figure is attached to the "Intervention" chapter.

6. Why did you use PNF and Vojta methods?

These are methods of proven effectiveness in the treatment of patients with spinal shape disorders. I had previously published papers using these methods in treatment (https://doi.org/10.13075/ijomeh.1896.01314) and wanted to continue these analyses.

7. [Line 128] Why was individualized therapy applied rather than a unified intervention?

In physiotherapy, it is very difficult to apply a uniform intervention to a large group of patients. They react very individually to its various forms. The use of a single intervention would expose some of the participants to potentially lower treatment effectiveness. Given the long duration of the experiment and the young age of the study participants, such an action would be unethical.

8. [Line 129] Is there a reason for the speed of 2 km/h in the measurement?

We wanted to test all patients under the same conditions. The speed of 2 km/h allows children of this age to walk calmly and freely, the closest to a free walk.

 9. I would recommend creating subheadings to separate the results. It will improve readability.

Thank you for this suggestion, the appropriate division has been applied.

Thank you again for your comments. We hope that the text in its current form will meet the requirements.

Reviewer 3 Report

I would like to thank the authors for the interesting research.

However, I have some important comments.

The purpose of the work is not entirely clear [rows 76-78]. The authors try to present the monitoring process or create criteria for the monitoring process. I suggest considering and specifying the research aim formulation.

For the justification of the research objective in the introduction, it is necessary to discuss the process of monitoring the changes in the efficiency of equilibrium reactions. 

In the section Statistical methods used, it is necessary to provide an abbreviation of analysed statistical values (eg SD etc.) [rows 157-171].

Conclusions require adjustment. Conclusions do not disclose the reached study aims. I think it is necessary to present monitoring criteria equilibrium reactions in children with postural disorders, in the course of the therapeutic process.

Sincerely.

Author Response

Dear Reviewer,
Thank you very much for your time and valuable comments. I will address them in turn below.

  1. The purpose of the work is not entirely clear [rows 76-78]. The authors try to present the monitoring process or create criteria for the monitoring process. I suggest considering and specifying the research aim formulation.

Thanks for the suggestion, the description of the goal has been expanded.

2. For the justification of the research objective in the introduction, it is necessary to discuss the process of monitoring the changes in the efficiency of equilibrium reactions. 

Thank you for your suggestion. The relevant paragraph has been added in the Introduction section.

3. In the section Statistical methods used, it is necessary to provide an abbreviation of analysed statistical values (eg SD etc.) [rows 157-171].

Thanks for the suggestion, added the relevant explanations.

4. Conclusions require adjustment. Conclusions do not disclose the reached study aims. I think it is necessary to present monitoring criteria equilibrium reactions in children with postural disorders, in the course of the therapeutic process

Thank you for this extremely important attention. The conclusions have been redrafted, which, together with the information added earlier, should form a coherent whole.

Thank you again for your comments. We hope that the text in its current form will meet the requirements.
